Potential role of cellular miRNAs in coronavirus-host interplay

Nersisyan Stepan snersisyan@hse.ru
http://orcid.org/0000-0003-3524-8586 Engibaryan Narek
Gorbonos Aleksandra
Kirdey Ksenia
http://orcid.org/0000-0002-5512-6757 Makhonin Alexey
Tonevitsky Alexander
Faculty of Biology and Biotechnology, HSE University , Moscow , Russia
Uversky Vladimir
Electronic publication date: 2020 Sep 14
Publication date: 2020
Volume: 8
Electronic Location ID: e9994
Received 2020 Jul 8; Accepted 2020 Aug 28
Copyright: © 2020 Nersisyan et al.
Copyright year: 2020
Copyright holder: Nersisyan et al.
License: This is an open access article distributed under the terms of the Creative Commons Attribution License, which permits unrestricted use, distribution, reproduction and adaptation in any medium and for any purpose provided that it is properly attributed. For attribution, the original author(s), title, publication source (PeerJ) and either DOI or URL of the article must be cited.
License URL: https://creativecommons.org/licenses/by/4.0/

Keywords: Coronavirus, miRNA, miR-21-3p, SARS-CoV-2, COVID-19, miR-195-5p, miR-16-5p, miR-3065-5p, miR-424-5p, miR-421

Funding: Russian Academic Excellence Project “5-100” The research was performed within the framework of the Basic Research Program at HSE University and funded by the Russian Academic Excellence Project “5-100”. The funders had no role in study design, data collection and analysis, decision to publish, or preparation of the manuscript.

==============================
Host miRNAs are known as important regulators of virus replication and pathogenesis. They can interact with various viruses through several possible mechanisms including direct binding of viral RNA. Identification of human miRNAs involved in coronavirus-host interplay becomes important due to the ongoing COVID-19 pandemic. In this article we performed computational prediction of high-confidence direct interactions between miRNAs and seven human coronavirus RNAs. As a result, we identified six miRNAs (miR-21-3p, miR-195-5p, miR-16-5p, miR-3065-5p, miR-424-5p and miR-421) with high binding probability across all analyzed viruses. Further bioinformatic analysis of binding sites revealed high conservativity of miRNA binding regions within RNAs of human coronaviruses and their strains. In order to discover the entire miRNA-virus interplay we further analyzed lungs miRNome of SARS-CoV infected mice using publicly available miRNA sequencing data. We found that miRNA miR-21-3p has the largest probability of binding the human coronavirus RNAs and being dramatically up-regulated in mouse lungs during infection induced by SARS-CoV.

Introduction

Coronavirus disease 2019 (COVID-19) caused by severe acute respiratory syndrome coronavirus 2 (SARS-CoV-2) acquired pandemic status on March 11, 2020, making a dramatic impact on the health of millions of people (Zhou et al., 2020a; Remuzzi & Remuzzi, 2020). Lung failure induced by the acute respiratory distress syndrome (ARDS) is the most common cause of death during viral infection (Xu et al., 2020).

MicroRNAs (miRNAs) are short (22 nucleotides in average) non-coding RNAs which appear to regulate at least one-third of all human protein-coding genes (Nilsen, 2007). Namely, in association with a set of proteins miRNA forms an RNA-induced silencing complex (RISC) and binds 3′-UTR of a target mRNA. The latter promotes translation repression or even mRNA degradation (Carthew & Sontheimer, 2009). Multiple works suggest the critical function of miRNAs in the pathogenesis of various human diseases. Thus, alteration of miRNAs expression is observed during different types of cancer (Di Leva, Garofalo & Croce, 2014; Shkurnikov et al., 2019), cardiovascular (Schulte et al., 2015; Nouraee & Mowla, 2015) and neurological diseases (Leidinger et al., 2013; Christensen & Schratt, 2009). Other studies have suggested that miRNAs can also participate in intercellular communication (Turchinovich, Drapkina & Tonevitsky, 2019; Baranova, Maltseva & Tonevitsky, 2019).

There are numerous reports consistently demonstrating the role of miRNAs in viral infections. One of the research directions deals with miRNAs which can target viral RNAs. Since RNA of a single-stranded RNA virus (ssRNA virus) is not structurally distinguishable from host mRNA, there are no barriers for miRNA to bind it. In contrast to conventional binding to 3′-UTR of target mRNA, host miRNAs often bind to the coding region or 5′-UTR of viral RNA (Bruscella et al., 2017). Besides translational repression, such interactions can also enhance viral replication or purposefully alter the amount of free miRNAs in a cell (Trobaugh & Klimstra, 2017). For example, miR-122 can bind to 5′-UTR of the hepatitis C virus (HCV) RNA which increases RNA stability and viral replication since it becomes protected from a host exonuclease activity (Shimakami et al., 2012). Another report contains evidence that miR-17 binding sites on RNA of bovine viral diarrhea virus (BVDV) seek to decrease level of free miR-17 in the cell, therefore mediating expression of miRNA targets (Scheel et al., 2016).

Other research groups focused on miRNAs altering their expression in response to the viral infection. Specifically, Liu et al. (2017) showed that proteins of avian influenza virus H5N1 cause upregulation of miR-200c-3p in the lungs. This miRNA targets 3′-UTR of ACE2 mRNA, therefore decreasing its expression. On the other hand, it was shown that decrease in ACE2 expression is critical in ARDS pathogenesis (Imai et al., 2005). Therefore, H5N1 virus promotes miRNA-mediated ACE2 silencing to induce ARDS. Recent reports suggest several other host miRNAs which can potentially regulate ACE2 and TMPRSS2 expression which can be also important during SARS-CoV-2 infection due to crucial role of these enzymes for virus cell entry (Nersisyan et al., 2020). Another example was given by Choi et al. (2014), who studied miRNAs altering their expression during influenza A virus infection. It was proved that several miRNAs which play an important role in cellular processes, including immune response and cell death, exhibited significant expression differences in infected mice. In the same article authors show that treatment with the respective anti-miRNAs demonstrates an effective therapeutic action.

In a recent article, Fulzele et al. (2020) found hundreds of miRNAs which can potentially bind to SARS-CoV-2 RNA as well as the RNA of the highly similar SARS-CoV coronavirus. However, this large miRNA list should be narrowed to find high-confidence interactions which should be further experimentally validated. In this work we hypothesized that there can be miRNA-mediated virus-host interplay mechanisms common for several human coronaviruses. For that purpose, we used bioinformatic tools to predict miRNA binding sites within human coronavirus RNAs including ones inducing severe acute respiratory syndrome (SARS-CoV-2, SARS-CoV and MERS-CoV) as well as other human coronaviruses causing common cold, namely, HCoV-OC43, HCoV-NL63, HCoV-HKU1 and HCoV-229E. To find and explore more complex regulatory mechanisms, we also analyzed miRNome of mouse lungs during SARS-CoV infection to find miRNAs whose expression was significantly altered upon viral infection.

Materials and Methods

Prediction of miRNA binding sites

To find miRNAs which can bind to viral RNAs we used miRDB v6.0 (Chen & Wang, 2020) and TargetScan v7.2 (Agarwal et al., 2015). Target predictions were filtered according to their miRDB target scores, threshold value was set to 75 as in for example, (Nakano et al., 2019; Zhuang, Bai & Liu, 2019). Viral genomes and their annotations were downloaded from the NCBI Virus (Hatcher et al., 2017) under the following accession numbers:NC_045512.2 (SARS-CoV-2);

NC_004718.3 (SARS-CoV);

NC_019843.3 (MERS-CoV);

NC_006213.1 (HCoV-OC43);

NC_005831.2 (HCoV-NL63);

NC_006577.2 (HCoV-HKU1);

NC_002645.1 (HCoV-229E).

To analyze miRNA-mRNA interactions, we also used miRTarBase v8 (Huang et al., 2020). DIANA-miRPath v3.0 was employed for KEGG pathway analysis (Vlachos et al., 2015).

RNA sequencing data and differential expression analysis

MiRNA sequencing (miRNA-seq) data from The Cancer Genome Atlas Lung Adenocarcinoma (TCGA-LUAD) project (Collisson et al., 2014) was used to quantify miRNA expression in the human lungs. Specifically, the said data was downloaded from GDC Data Portal (https://portal.gdc.cancer.gov/) and included miRNA expression table with its columns correspond to 46 normal lung tissues and rows associated with miRNAs (note that only small fraction of TCGA-LUAD cancer samples had analyzed matched normal tissues). We used log2-transformed Reads Per Million mapped reads (RPM) as a miRNA expression unit.

Two miRNA-seq datasets, GSE36971 (Peng et al., 2011) and GSE90624 (Morales et al., 2017), were used to analyze miRNome of SARS-CoV infected mouse lungs (RNA was extracted from homogenized whole lung lobes). Raw FASTQ files were downloaded from the Sequence Read Archive (Leinonen, Sugawara & Shumway, 2011). Adapters were trimmed via Cutadapt 2.10 (Martin, 2011), miRNA expression was quantified by miRDeep2 (Friedländer et al., 2012) using GRCm38.p6 mouse genome (release M25) from GENCODE (Frankish et al., 2019) and miRBase 22.1 (Kozomara, Birgaoanu & Griffiths-Jones, 2019). Gene expression profile of SARS-CoV infected mouse lungs was downloaded in form of count matrix from the Gene Expression Omnibus (GEO) (Barrett et al., 2013) under GSE52405 accession number (Josset et al., 2014). Differential expression analysis was performed with DESeq2 (Love, Huber & Anders, 2014). The results were filtered using 0.05 threshold on adjusted p-value and 1.5 on fold change (linear scale).

Sequence alignment

Multiple Sequence Alignment (MSA) of viral genomic sequences was done using Kalign 2.04 (Lassmann, Frings & Sonnhammer, 2009). Two MSA series were performed. In the first one we aligned seven human coronavirus genomes. In the second one different coronavirus strains were aligned for each of analyzed viruses. All genomes available on the NCBI Virus were used for SARS-CoV, MERS-CoV, HCoV-OC43, HCoV-NL63, HCoV-HKU1 and HCoV-229E (263, 253, 139, 58, 39 and 28 genomes, respectively). For SARS-CoV-2 thousand genomes were randomly selected to preserve the percentage of samples from each country. GISAID clade annotation (Elbe & Buckland-Merrett, 2017) was obtained for 956 SARS-CoV-2 genomes (annotation was missing for other genomes). For each virus we established the mapping between alignment and genomic coordinates. With the use of this mapping, miRNA seed region binding positions within viral RNAs were placed on the alignment.

Data and code availability

All code was written in Python three programming language with extensive use of NumPy (Van Der Walt, Colbert & Varoquaux, 2011) and Pandas (McKinney, 2010) modules. Statistical analysis was performed using the SciPy stats (Virtanen et al., 2020), plots were constructed using the Seaborn and Matplotlib (Hunter, 2007). MSA was visualized using Unipro UGENE (Okonechnikov et al., 2012). All used data and source codes are available on GitHub (https://github.com/s-a-nersisyan/host-miRNAs-vs-coronaviruses).

Results

Human coronavirus RNAs have numerous common host miRNA binding sites

To identify human miRNAs that may bind to RNAs of human coronaviruses we used two classical miRNA target prediction tools: miRDB and TargetScan. TargetScan results can be ranked with different seed-region binding types while miRDB results can be ranked with so-called “target score” associated with the probability of successful binding. Interestingly, for each of viruses TargetScan predicted 2–3 times higher number of miRNAs, while 80–85% miRNAs predicted by miRDB were predicted by TargetScan too (for the summary on the number of miRNAs predicted for each of viral genomes see Table S1).

We made a list of 19 miRNAs potentially targeting multiple viral RNAs by selecting miRNAs with miRDB target scores greater than 75 in all analyzed viruses (Table S2). For further analysis, we selected only high confidence miRNAs according to miRBase, namely, hsa-miR-21-3p, hsa-miR-195-5p, hsa-miR-16-5p, hsa-miR-3065-5p, hsa-miR-424-5p and hsa-miR-421. According to TargetScan, all “guide” strand miRNAs except hsa-miR-3065-5p were conserved among species including miR-16-5p/195-5p/424-5p family with the shared seed sequence. Despite being a “passenger” strand, hsa-miR-21-3p was shown to be functionally active and conserved over the mammalian evolution (Báez-Vega et al., 2016; Lo, Tsai & Chen, 2013). For target scores of selected miRNAs as well as corresponding hierarchical clustering of viruses see Fig. 1A. As it can be seen, such clustering grouped together SARS-CoV and SARS-CoV-2 as well as HCoV-229E and HCoV-NL63 which can be also observed when clustering is performed based on viral genomic sequences similarity (Zhou et al., 2020b).

Figure 1 miRNAs with the highest target scores.

(A) Hierarchical clustering of coronaviruses based on miRDB target scores. Rows are sorted according to the mean target score. (B) Expression distribution of miRNAs in human lungs. Diamonds on the boxplot represent outliers.

Six identified miRNAs showed similar functional patterns. Namely, KEGG pathway analysis of experimentally validated target genes revealed 54 enriched terms including pathways involved in pathogenesis of lung and several other cancers, viral infections as well as signaling pathways such as p53, TGF-β and FoxO (see Table S3). In order to assess which of these miRNAs could demonstrate activity in human lungs, we analyzed miRNA-seq data from TCGA-LUAD project. Two of the said miRNAs demonstrated relatively high expression (see Fig. 1B). Specifically, hsa-miR-21-3p and hsa-miR-16-5p corresponded to top-5% of highly expressed miRNAs according to their mean expression level taken across all samples.

Viral binding sites of miRNAs are conserved across different coronaviruses and their strains

Each of identified miRNAs had dozens of binding regions within analyzed viral RNAs (see Table S4). Interestingly, the peak number of hsa-miR-16-5p/195-5p/424-5p binding positions fell on SARS-CoV and SARS-CoV-2, while for other miRNAs the most enriched virus was HCoV-NL63. To go deeper and analyze mutual arrangement of these sites we performed multiple sequence alignment on seven analyzed genomes and mapped the predicted miRNA binding positions from individual genomes to the obtained alignment. Further, for each binding site mapped to the alignment we calculated a number of viruses sharing that particular miRNA binding site. Positions common for two or more viruses were utilized in the downstream analysis.

In general, different miRNAs demonstrated dissimilar patterns of viral binding regions (for summary information see Table 1). In particular, hsa-miR-21-3p and hsa-miR-421 had positions on the alignment specific to six out of seven considered coronaviruses (see Fig. 2). Two most enriched binding positions of hsa-miR-16-5p/195-5p/424-5p family were common for five viruses, while maximum number of viruses sharing binding regions of hsa-miR-3065-5p was equal to three. Interestingly, most binding sites obtained for all considered miRNAs were found within nonstructural proteins located in polyprotein 1ab coding region (89%), about 8% of positions were located within spike protein while the rest was spread over N and M proteins. Detailed information is given in Table S5.

Table 1 Number of common miRNA binding sites on coronavirus RNAs.

Column names refer to the number of viruses sharing a miRNA binding region.

	2	3	4	5	6	Total	
hsa-miR-21-3p	24	6	2	1	1	34	
hsa-miR-16-5p/195-5p/424-5p	16	5	0	2	0	23	
hsa-miR-3065-5p	41	11	0	0	0	52	
hsa-miR-421	34	3	2	2	1	42	

Figure 2 Shared binding sites of hsa-miR-21-3p and hsa-miR-421 on human coronavirus RNAs.

(A) hsa-miR-21-3p. (B) hsa-miR-421.

To group coronaviruses based on the probability of sharing common miRNA binding positions, we calculated the number of matching positions in the alignment for each miRNA and pair of viruses (see Fig. S1). Then, per each miRNA this data was normalized by the overall number of binding positions shared by two or more viruses, and used as a distance matrix for hierarchical clustering (see Fig. 3). Interestingly, for majority of miRNAs such clustering was completely similar to that of viruses based on their genomic sequence similarity (Zhou et al., 2020b).

Figure 3 Hierarchical clustering of human coronaviruses based on the number of shared binding sites.

(A) hsa-miR-21-3p. (B) hsa-miR-16-5p/195-5p/424-5p. (C) hsa-miR-3065-5p. (D) hsa-miR-421.

In order to assess conservativity of miRNA binding regions across viral strains, we performed multiple sequence alignment of available viral genomes per each human coronavirus independently. The results revealed high conservativity of these regions: 59–98% of binding sites within coronavirus RNAs had no mutation, while mean of average mutation rates (i.e., number of mutations normalized by region length and number of strains) across all viruses varied from 0.3% to 0.7% (see Table S4). Interestingly, there were no mutations for each of viruses in aforementioned hsa-miR-21-3p binding site shared by six coronaviruses, while HCoV-OC43 mutation rate was equal to 1% in the similar hsa-miR-421 site. Exact values of average mutation rates are given in Table S6. Finally, we assessed mutation rates within seven SARS-CoV-2 clades introduced by GISAID. Binding sites of hsa-miR-21-3p and hsa-miR-421 had mismatches only within GH and S clades, sites of hsa-miR-3065-5p were mutated in GH and GR clades, while binding regions of hsa-miR-16-5p/195-5p/424-5p family showed mutations in all clades except O (see Table 2).

Table 2 Mean of average mutation rates in miRNA binding sites across SARS-CoV-2 clades.

	G	GH (%)	GR	L	O	S	V	
hsa-miR-21-3p	0	0.007	0	0	0	0.03%	0	
hsa-miR-16-5p/195-5p/424-5p	0.01%	0.03	0.01%	0.02%	0	0.05%	0.08%	
hsa-miR-3065-5p	0	0.02	0.01%	0	0	0	0	
hsa-miR-421	0	0.003	0	0	0	0.01%	0	

miR-21 and its target genes exhibit significant expression alteration in mouse lungs during SARS-CoV infection

To further explore a potential interplay between host miRNAs and coronaviruses, we hypothesized that some of miRNAs predicted to bind viral RNAs can have altered expression during the infection. In order to verify this hypothesis, we analyzed two publicly available miRNA-seq datasets of mouse lungs during SARS-CoV infection. The first dataset (GSE36971) included data derived from four mouse strains infected by SARS-CoV and four corresponding control mice. The second dataset (GSE90624) comprised three infected and four control mice.

Differential expression analysis revealed 19 miRNAs in the first dataset and 21 in the second dataset where expression change during infection was statistically significant (see Table S7). Six miRNAs were differentially expressed in both datasets, five of them had matched fold change signs, namely, were overexpressed in infected mice (see Fig. 4). This was a statistically significant overlap since an estimate of the probability for 19- and 21-element random miRNA sets having five or more common elements was equal to 4.07 × 10−7 (hypergeometric test). Surprisingly, miR-21a-3p which we previously identified as a potential regulator of all analyzed coronavirus genomes with one of the highest scores exhibited 8.3-fold increase (adjusted p = 5.72 × 10−35) and 11-fold increase (adjusted p = 5.77 × 10−11) during SARS-CoV infection in the first and the second datasets, respectively.

Figure 4 Differentially expressed mouse lung miRNAs.

(A) GSE36971. (B) GSE90624. Differential expression analysis was performed using DESeq2. Note that counts were normalized independently per each dataset. Thus, presented values should not be directly compared across (A) and (B). Vertical lines on the bar top indicate 95% confidence intervals.

Expression of mmu-miR-21a-5p (the opposite “guide” strand of the same hairpin) was also increased in the infected group (2.8- and 3.2-folds, respectively). The latter had a particular importance since mmu-miR-21a-5p was highly expressed in mice during both experiments. Namely, according to its mean expression across all samples it was 4th and 38th out of 2,302 in the first and the second datasets, respectively. Thus, significant expression change of this miRNA can dramatically affect expression of its target genes.

In order to capture aberrant expression of miRNA target genes during infection, we analyzed RNA sequencing (RNA-seq) data of eight SARS-CoV infected mice strains published by the same group of authors as in the first miRNA-seq dataset (GSE52405). Two strategies were pursued to generate a list of miRNA targets. Namely, we used target prediction tools described in the previous section as well as literature-curated miRTarBase database.

First, we took genes predicted both by miRDB and TargetScan with miRDB target score greater than 75. Additionally, we thresholded this list using top-10% predictions based on TargetScan’s cumulative weighted context++ score. A significant fraction of mmu-miR-21a-5p target genes were down-regulated during the infection. Namely, 6 out of 24 considered genes demonstrated significant decrease in expression (hypergeometric test p = 7.6 × 10−3). For four other miRNAs, there was no statistical significance on the number of down-regulated target genes. The situation was quite different for interactions enlisted in miRTarBase. Thus, 2 out of 2 mmu-miR-21a-3p target genes (Snca and Reck) were down-regulated (hypergeometric test p = 5.7 × 10−3), while only 6 out of 37 mmu-miR-21a-5p target genes exhibited expression decrease (hypergeometric test p = 0.057). As in the previous case, no significant number of down-regulated target genes was observed for other miRNAs.

Discussion

In this article we identified several cellular miRNAs (miR-21-3p, miR-16-5p/195-5p/424-5p, miR-3065-5p and miR-421) potentially regulating all human coronaviruses via direct binding to viral RNAs. Moreover, aside from virus specific binding sites we identified genomic positions which can serve as conserved targets for putative miRNAs. As one can expect, viruses with high genomic similarity such as SARS-CoV-2/SARS-CoV or HCoV-229E/HCoV-NL63 had higher number of shared binding sites. Similar computational approach to discover direct miRNA-virus interactions was already employed by Fulzele et al. (2020). Namely, using miRDB researchers predicted miRNAs targeting RNAs of SARS-CoV-2 and SARS-CoV. Despite large intersection of predicted miRNA sets, in the present article we focused on miRNAs targeting as much human coronavirus RNAs as possible, which resulted in different lists of “top” miRNAs.

Several hypotheses can be put forward to explain biological motivation of direct host miRNA-virus interactions. At first sight, one can think about host miRNA-mediated immune response to the viral infection. For example, translation of human T cell leukemia virus type I (HTLV-1) is inhibited by miR-28-3p activity (Bai & Nicot, 2015). However, our results suggest that binding sites of identified miRNAs are actually conserved across human coronaviruses. Thus, viruses can purposefully accumulate host miRNA binding sites to slow down their own replication rate in order to evade fast detection and elimination by the immune system. Such behavior was reported for example, in the case of eastern equine encephalitis virus (EEEV) (Trobaugh et al., 2014). Authors reported that hematopoietic-cell-specific miRNA miR-142-3p directly binds viral RNA which limits the replication of virus thereby suppressing innate immunity. The latter was shown to be crucial in the virus infection pathogenesis.

Functional activity of identified miRNAs was already referred to in the context of viral infections. Thus, it was proved that miR-21-3p regulates the replication of influenza A virus (IAV) (Xia et al., 2018). Namely, hsa-miR-21-3p targeting 3′-UTR of HDAC8 gene was shown to be down-regulated during IAV infection of human alveolar epithelial cell line A549 using both miRNA microarray and quantitative PCR analysis. Consecutive increase in the HDAC8 expression was shown to promote viral replication. Another report highlights the role of miR-16-5p in pathogenesis of Enterovirus 71 (EV71) infection (Zheng et al., 2017). In particular, authors validated EV71-induced expression of miR-16-5p and found that this miRNA can inhibit EV71 replication in vitro and in vivo by targeting CCNE1 and CCND1 genes.

Remarkably, we found that expression of miR-21-3p in mice lungs exhibits a 8-fold increase upon SARS-CoV infection. Interestingly, miR-21-5p (the “guide” strand of the same pre-miRNA hairpin) demonstrated only a 3-fold increase in expression. To explain this phenomena of non-symmetrical expression increase, we hypothesize that binding to he viral genome saves star miRNA miR-21-3p from degradation after unsuccessful attempt of AGO2 loading. A similar mechanism was already mentioned in several articles. Namely, Janas et al. (2012) demonstrated that Ago-free miRNAs can escape degradation by forming Ago-free miRNA-mRNA duplex. Another concept was named as target RNA directed miRNA degradation (TDMD) and consists of the fact that highly complementary target RNA can trigger miRNA degradation by a mechanism involving nucleotide addition and exonucleolytic degradation (De la Mata et al., 2015; Zhang et al., 2019). Thus, non-proportional up-regulation of miR-21 arms can be indirect evidence that miR-21-3p directly targets the viral RNA or that the miR-21-5p is being actively degraded during target mRNA binding.

Conclusions

Several miRNAs having potential of direct interactions with human coronaviruses were discovered in this article. While a majority of them were virus-specific, some miRNAs were shown to target all analyzed viral RNAs. Exploration of publicly available miRNomic data of SARS-CoV infected mice lungs revealed that one of these miRNAs (miR-21-3p) demonstrated a dramatic expression increase upon infection. Taking into account high structural similarity of SARS-CoV and SARS-CoV-2 including common miR-21-3p binding sites as well as the fact that this miRNA is also expressed in human lungs, the obtained results open new opportunities in understanding COVID-19 pathogenesis and consecutive development of therapeutic approaches.

Supplemental Information

Supplemental Information 1 Number of shared binding sites for each pair of human coronaviruses.

(A) hsa-miR-21-3p. (B) hsa-miR-16-5p/195-5p/424-5p. (C) hsa-miR-3065-5p. (D) hsa-miR-421.

Click here for additional data file.

Supplemental Information 2 Number of miRNAs predicted to bind viral RNAs.

Click here for additional data file.

Supplemental Information 3 miRNAs with target score greater than 75 for all viruses.

Values in cells of the table refer to miRDB target scores.

Click here for additional data file.

Supplemental Information 4 KEGG pathway analysis of six miRNA targets.

Click here for additional data file.

Supplemental Information 5 Number of miRNA binding sites within coronavirus RNAs.

Seed region binding types are named according to TargetScan.

Click here for additional data file.

Supplemental Information 6 Binding sites of miRNAs shared by two or more viruses.

Vertical bars on the alignment denote regions complementary to seed regions.

Click here for additional data file.

Supplemental Information 7 Conservativity of miRNA binding sites.

Click here for additional data file.

Supplemental Information 8 The results of miRNA differential expression analysis in SARS-CoV infected mice lungs.

Click here for additional data file.

The authors thank Dr. Maxim Shkurnikov for valuable comments and discussions.

Additional Information and Declarations

Competing Interests

Author Contributions

Data Availability

The authors declare that they have no competing interests.

Stepan Nersisyan conceived and designed the experiments, performed the experiments, analyzed the data, prepared figures and/or tables, authored or reviewed drafts of the paper, and approved the final draft.

Narek Engibaryan performed the experiments, analyzed the data, authored or reviewed drafts of the paper, and approved the final draft.

Aleksandra Gorbonos performed the experiments, analyzed the data, prepared figures and/or tables, authored or reviewed drafts of the paper, and approved the final draft.

Ksenia Kirdey performed the experiments, analyzed the data, authored or reviewed drafts of the paper, and approved the final draft.

Alexey Makhonin performed the experiments, analyzed the data, authored or reviewed drafts of the paper, and approved the final draft.

Alexander Tonevitsky conceived and designed the experiments, analyzed the data, authored or reviewed drafts of the paper, and approved the final draft.

The following information was supplied regarding data availability:

Raw data are available in the Supplementary Tables. Source code and other technical data are available on GitHub:

https://github.com/s-a-nersisyan/host-miRNAs-vs-coronaviruses.

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
