# Peer review of "Potential role of cellular miRNAs in coronavirus-host interplay"

_PeerJ, doi:10.7717/peerj.9994_

## Round 0.1 · original submission · Major Revisions

As you can see, all the reviewers provided rather positive comments, but also some revision was requested. Please address all the critiques and revise your manuscript accordingly.

·

Basic reporting

The manuscript reads well. The topic is highly relevant to the current pandemic and sufficient background and literature were searched for the current work.

Experimental design

The experiments have been well designed and conducted. However, I have a few questions regarding the experiments and potentially, some other data could be provided, the manuscript will be even better.
1. As authors suggest Virus infection could soak up the miRNA to prevent its function. Is it possible to show the potential targets of this miRNA in human lungs mRNA, which could be altered by the miRNA-21-3p
2. GISAID website divided the SARSCov2 genome into five clades, S, Gr, GH, G, S, and V. Is possible to test whether miRNA-21-3p binding sequences remain intact in all of these clades or its potential binding will be limited to viruses of certain clades only.

Validity of the findings

The manuscript dealt with the use of various bioinformatic tools and they seem appropriate.

Reviewer 2 ·

Basic reporting

The manuscript “The potential role of miR-21-3p in coronavirus-host interplay " is interesting, the aim is clear and the methods seem appropriate. In this work, authors performed computational prediction of high-confidence direct interactions between miRNAs and seven human coronavirus RNAs.

Experimental design

At this unprecedented time, researchers are performing number of hypothesized based studies. I completely support such studies but this study has one big flaws. Authors gave too much importance to one miR-21-3p. This study is good and made some valid points but focusing on one miRNA made study looks bad.

Validity of the findings

1. I recommend changing title (not focusing on one miRNA)
2. Change discussion and mention about other miRNAs like 16-5p, 195-5p
3. Delete portion of miR-21-3p and 5p discussion part. It looks great on paper about discussing 5p and 3p…unless in-vivo or in-vitro studies. We perform such studies and outcomes are disappointing.
4. Make this manuscript generalize, don’t focus on one miRNA.
5. Mice lung data is ok but in reality miRNA targets are different in human and mice.

Reviewer 3 ·

Basic reporting

Major:
1. Functionality of miR-21-3p in different viral disease conditions are missing in the introduction.
2. Which part of the lungs was used for this study?
3. Is this miRNA (21-3p) cross species conserved?
4. Statistical analysis for Fig. 2 is missing.
5. It will be interesting to know about the other targets of miR-21-3p.
6. A table summarizing the fold change and targets of miR-21-3p in different disease conditions will be interesting to note.

Minor:
1. English language editing required.

Experimental design

No comments.

Validity of the findings

Statistical analysis for Fig. 2 is missing.

Additional comments

The review article by Nersisyan et al. entitled “The potential role of miR-21-3p in coronavirus-host interplay” discusses the novel role of miRNA miR-21-3p in binding to the human coronavirus RNAs and its role in SARS-CoV infection. The article is timely, descriptive, interesting, however, there are few concerns which the author needs to address:

Reviewer 4 ·

Basic reporting

No comment

Experimental design

No comment

Validity of the findings

No comment

Additional comments

The authors showed the potential role of miR-21-3p in coronavirus-host interplay via bioinformatic analysis. Since the idea is of interest, there are some points that need to be addressed to improve the quality of study.

There still have some grammatical errors and typos. The authors should re-check and revise carefully.

One of the important concerns is that the authors have not compared the performance results to the previous works on mRNA binding site prediction. Even currently no study focused on coronavirus specifically but the authors could also compare to the general one.

There are many mRNA samples in TCGA-LUAD, why did the authors only select 46 tissues? Any criteria for this?

Why did the authors select genes with target score > 75?

---

## Round 0.2 · accepted · Accept

Since all the critical issues were addressed and manuscript was amended accordingly, I am glad to accept revised version.

Reviewer 4 ·

Basic reporting

No comment

Experimental design

No comment

Validity of the findings

No comment

Additional comments

My previous comments have been addressed well.